# miR-122 removal in the liver activates imprinted microRNAs and enables more effective microRNA-mediated gene repression

Paul N. Valdmanis[1,2,3], Hak Kyun Kim[1,2], Kirk Chu[1,2], Feijie Zhang[1,2], Jianpeng Xu[1,2], Elizabeth M. Munding[1,2], Jia Shen[1,2] & Mark A. Kay[1,2]

miR-122 is a highly expressed liver microRNA that is activated perinatally and aids in regulating cholesterol metabolism and promoting terminal differentiation of hepatocytes. Disrupting expression of miR-122 can re-activate embryo-expressed adult-silenced genes, ultimately leading to the development of hepatocellular carcinoma (HCC). Here we interrogate the liver transcriptome at various time points after genomic excision of miR-122 to determine the cellular consequences leading to oncogenesis. Loss of miR-122 leads to specific and progressive increases in expression of imprinted clusters of microRNAs and mRNA transcripts at the *Igf2* and *Dlk1-Dio3* loci that could be curbed by re-introduction of exogenous miR-122. mRNA targets of other abundant hepatic microRNAs are functionally repressed leading to widespread hepatic transcriptional de-regulation. Together, this reveals a transcriptomic framework for the hepatic response to loss of miR-122 and the outcome on other microRNAs and their cognate gene targets.

[1] Department of Pediatrics, Stanford University, Stanford, 94305 CA, USA. [2] Department of Genetics, Stanford University, Stanford, 94305 CA, USA. [3] Division of Medical Genetics, University of Washington School of Medicine, Seattle, 98195 WA, USA. Correspondence and requests for materials should be addressed to P.N.V. (email: paulnv@uw.edu) or to M.A.K. (email: markay@stanford.edu)

MicroRNAs mediate site-specific repression of mRNA targets through complementarity base pairing. The number of microRNA transcripts in mammals is about an order of magnitude less than the number of genes and their corresponding mRNAs. Consequently, while each mRNA species constitutes a small fraction of the total miRNAs in a cell, certain microRNAs have a much greater representation of the total microRNA pool and can exert a greater influence on their cellular microRNA counterparts. This discrepancy is especially notable in the liver where one major microRNA, miR-122, accounts for the bulk (~70%) of liver microRNA read counts, which is in contrast to abundant mRNAs such as Transferrin (Trf) that represent no more than 1–2% of total mRNA reads[1]. The liver represents an ideal organ for studying microRNA biology given its accessibility and ease of transduction of the vast majority of hepatocytes by recombinant adeno-associated virus (rAAV) vectors[2]. Our work evaluating the effect that high small hairpin RNA (shRNA) expression has on liver microRNA levels and particularly the various abundant isoforms of miR-122-5p has established guidelines for identifying and interpreting other situations where miR-122 isoforms are altered[3]. miR-122-5p is initially synthesized as a 22-nt species which is further mono-adenylated[4] or mono-uridylated[3] to a 23-nt species or trimmed by one nucleotide to a 21-nt isoform. The 22-nt isoform of miR-122 is specifically competed by shRNAs.

Several studies have sought to address the quantification of microRNAs and their role in reduction of target mRNAs[1,5–7]. The contribution of various inhibitors of abundant microRNAs, especially miR-122, has also been evaluated[8–10], though this is generally restricted to the existing targets of the microRNA in question. It is still unclear how the absence of miR-122 affects the relative abundance and function of other microRNAs in the mouse liver. Upon elimination of newly synthesized miR-122, we established the kinetics of miR-122 loss, identified a pathway of processing intermediates for mature miR-122-5p, determined the hepatic microRNAs that are activated in response to miR-122 loss and identified the consequences to mRNAs that are targets of other highly expressed microRNAs in the liver. This provides transcriptional responses in the liver leading to HCC. Collectively, this establishes a quantitative framework for the function of a highly abundant microRNA in the context of cellular mRNAs and microRNA counterparts.

## Results

**miR-122 progressively declines after genomic excision**. To identify how quickly miR-122 is degraded, we administered $1 \times 10^{12}$ vector genomes of rAAV8-Ef1a-Cre in adult mice with Lox-P sites surrounding pre-miR-122 to remove miR-122 and harvested liver samples at various time points in as we have previously performed[3]. This revealed a consistent drop in miR-122 expression relative to the expression of all microRNAs (Fig. 1a, Supplementary Data 1). This effect was quite pronounced, commencing as early as 1−2 days post-transduction (Fig. 1a).

**miR-122-5p degradation occurs through poly-uridylation**. In our previous data, we found that the 22-nt isoform of miR-122-5p is the first to be generated and the first to be degraded after excision of miR-122 [3]. We could extend this data to note that the 22-nt to 21-nt isoform ratio of miR-122-5p consistently drops after miR-122 removal (Fig. 1b), confirmed by small RNA northern blot quantification (Fig. 1c, d). Indeed, the levels of the 22-nt and 23-nt terminating in "U" had a more rapid decline relative to all microRNA read counts compared to the 21 nt and 23 nt terminating in "A" (Supplementary Figure 1a, b). This suggests that the stability of these two pairs of isoforms is biologically intertwined and is in equilibrium in hepatocytes. Several additional rare isoforms of miR-122-5p also were detected, such

as mono-uridylation or adenylation of a 23U or 23A isoform, that decreased in abundance in a manner proportional to other more common miR-122-5p species. Conversely, we found that mono- and poly-uridylation of the 21-nt isoform of miR-122-5p constituted a greater proportion of total miR-122-5p reads as the absolute level of miR-122 dropped. For instance, mono-uridylation of the 21-nt species (21+U) typically represents ~1.5% of total miR-122-5p reads but increased to 6.7% of miR-122 reads by 17 days post-Cre removal and accounted for more reads than the 22-nt isoform at this point (Fig. 1e). The levels of this 21+U species remained constant relative to all microRNA reads for the first 25 days strongly suggesting that this is a degradation intermediate that is continuously replenished until miR-122 is eliminated (Supplementary Figure 1a). It also indicates that our sequencing is likely normalized well between samples. Thus, miR-122-5p degradation proceeds through the nontemplated uridine addition of the 21-nt isoform (Fig. 1f).

**miR-122 regains expression after near-complete liver removal**. The nadir of miR-122 expression occurred at 45 days post-Cre delivery where it constituted <1% of miRNA reads. Surprisingly by ~100 days, miR-122 slightly rebounded in expression to ~4% of total microRNA reads and leads to ~40% of reads by 200 days (Fig. 1g). Based on our previous work, these reads represent new miR-122 because the isoform distribution profile and 22-nt to 21-nt ratios of miR-122 matches that of wild-type mice and we no longer observe the levels of the 21+U degradation byproducts to any greater extent than wild-type mice (Fig. 1h). DNA levels corresponding to the wild-type MiR122 locus also initially diminished rapidly and then partially rebounded at the 200-day timepoint (Supplementary Figure 1c). The new miR-122 species may reflect the selective growth advantage of hepatocytes that have escaped rAAV transduction and Cre removal, analogous to what was observed in $Dicer^{-/-}$ hepatocytes[11]. Notably, liver-specific $miR-122^{-/-}$ mice (Cre expressed under the Albumin promoter)[12] also retain some expression of miR-122 at ~6% of total liver microRNAs (Fig. 1g), again with an isoform distribution profile similar to wild-type mice (Fig. 1h). Therefore, some as yet unidentified conditions permit miR-122-expressing cells to re-emerge in the liver after near-complete ablation. Consequently, rare hepatocytes that retain expression of miR-122 may exert some selective advantage over their nonexpressing counterparts.

**miR-122-5p-22 nt is most abundant in the postnatal liver**. Given our understanding of the degradation process of miR-122, we wanted to establish if a converse pattern of miR-122 expression (and its isoforms) could be found in the developing mouse liver. Consequently, we performed small RNA-seq of liver samples commencing at embryo day E14.5 and including several early postnatal timepoints. miR-122 was quickly activated in development and already accounted for 39% percent of liver microRNA read counts at birth, reaching adult liver levels by 3 weeks of age (Fig. 2a). Not surprisingly, the 22-nt species constituted the bulk of reads, initially with a 22:21 nt ratio of 3.72 at birth (Fig. 2b). This abundance of the 22-nt species dropped consistently in the postnatal mouse and reached levels of the typical adult mouse by 4 weeks of age (Fig. 2c). Importantly, levels of the 21+U and 21+UU degradation byproducts only started to appear at later timepoints after birth (Fig. 2c). Together, our results confirm that miR-122 is first expressed late in embryonic development[13]. Our analysis of the miR-122 isoforms that begin to accumulate during fetal development are the direct converse of miR-122 isoform patterns after Cre-mediated miR-122 removal.

**Oncogenic microRNAs negatively correlate with miR-122 levels**. To evaluate whether eliminating miR-122 influences temporal expression of other microRNAs, we compared normalized read per million mapped microRNA counts for each

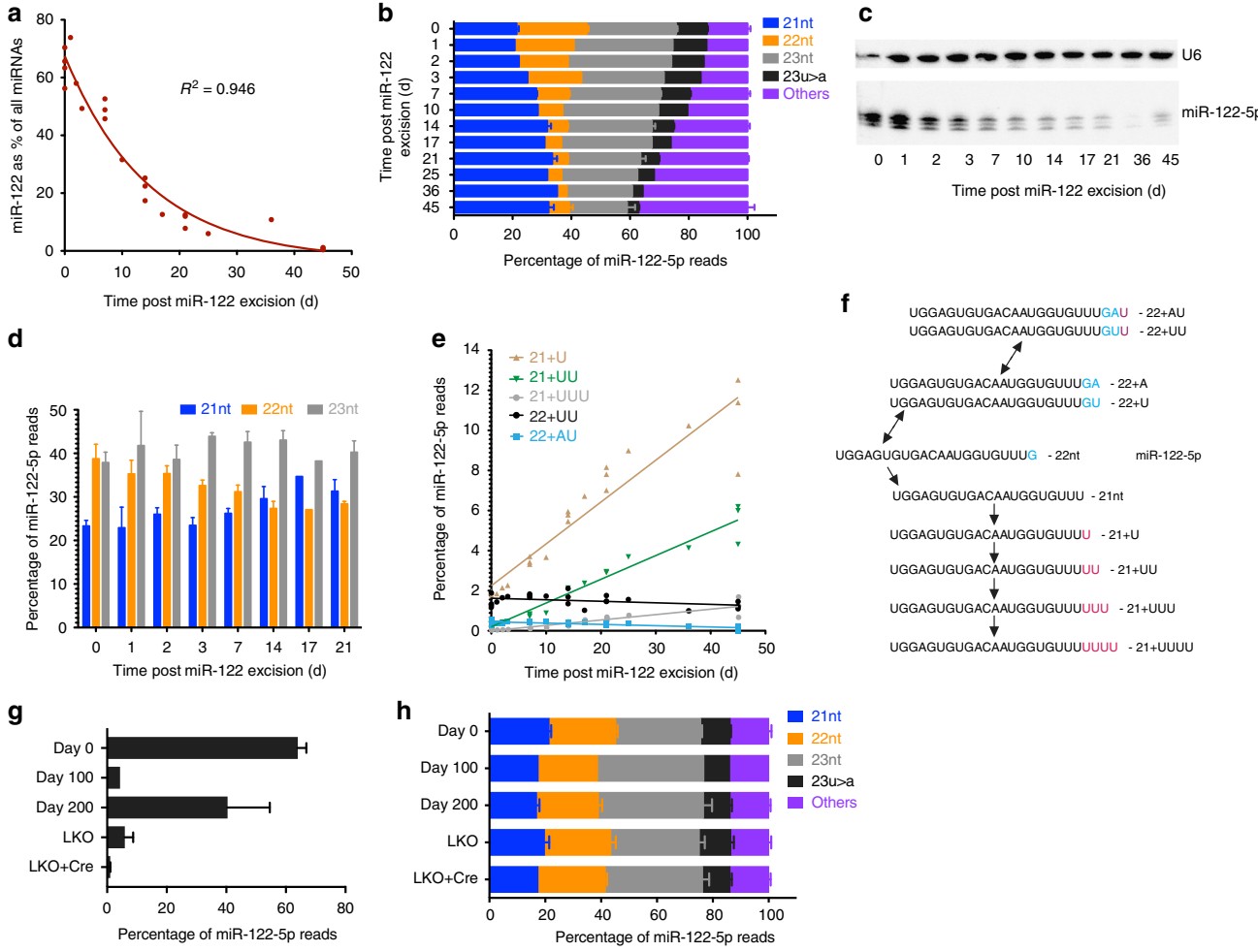

**Fig. 1** miR-122 has a consistent degradation pattern but remains expressed from a subset of cells. **a** Percentage of small RNA-seq reads that map to miR-122 relative to all other microRNAs in livers of miR-122 floxed mice at various time points after receiving $1 \times 10^{12}$ vector genomes of rAAV8-Cre ($n = 1$ mouse per point on the graph). **b** Relative abundance of main isoforms of miR-122-5p from small RNA-seq data ($n = 3$ mice for days 0, 7, 14, 21, and 45, error bars are s.e.m.; $n = 1$ for the remaining timepoints). **c** Small RNA northern blot of miR-122 expression in mouse livers at various timepoints after miR-122 excision. **d** Quantification of bands from **c**. **e** Proportion of rare miR-122-5p isoforms relative to all mapped microRNAs. **f** Schematic of the processing of miR-122-5p isoforms. **g** Percentage of miR-122 relative to all microRNAs at longer time periods after miR-122 removal and in mice expressing Cre under the Albumin promoter leading to a liver-specific knockout of miR-122 (LKO) ($n = 3$ for days 0 and 200, $n = 2$ for LKO and LKO + Cre and $n = 1$ for day 100). **h** miR-122-5p isoform distributions of samples from **g**

microRNA after excluding counts that map to miR-122. Under these parameters, as expected, the vast majority of microRNAs did not change in relative expression (Supplementary Figure 2). However, several microRNAs present in imprinted loci became markedly activated over time including miR-483 and miR-675 at the *Igf2* locus and nearly all microRNAs on the *Dlk1-Dio3* locus on mouse chromosome 12qF1, represented by miR-127 and miR-376a (Fig. 3a, b). Activation of microRNAs at the *chr12qF1* locus alone has been implicated in the development of hepatocellular carcinoma (HCC) in several contexts[14,15] including in miR-122 knockout mice[16] and is a signature we have also found in *Kras*[G12D]-induced lung adenocarcinoma[17], though notably the mouse lung cancer samples did not have activated miR-483 or miR-675 expression. Consistently increased expression of the combined *chr12qF1* locus microRNAs (but not a control micro-RNA, miR-21) is observed when comparing microRNA read counts from small RNA-seq of miR-122[−/−] mice relative to their age-matched wild-type counterparts (Fig. 3c, d). Delivery of miR-122 by rAAV vectors reduces imprinted microRNA levels at the chr12qF1 locus in miR-122 knockout mice (Fig. 3d) and may explain the reduction in tumor incidence when miR-122 is administered by hydrodynamic tail vein injection to miR-122[−/−]

mice[18]. Levels of *chr12qF1* and *Igf2* locus microRNAs were initially high in developmental stages and quickly decreased in relative abundance after birth (Fig. 3e). When analyzing normalized small RNA-seq data[3] from mice with and without miR-122[−/−] (after removing read counts that map to miR-122), *chr12qF1* and *Igf2* locus microRNAs are notably activated (Fig. 3f), which matches the levels of *chr12qF1* and *Igf2* microRNA levels in *Mir122* knockout mice (Fig. 3g). The expression of *chr12qF1* and *Igf2* locus microRNAs continues to increase even after the 100-day timepoint when miR-122 becomes re-expressed, though they begin to taper off. Immunohistochemistry of mouse liver sections 200 days after rAAV-Cre indicates marked fibrosis as has been described for miR-122 knockout mice (Supplementary Figure 3)[12,18]. Together this suggests a mechanism connecting miR-122 loss with microRNAs implicated in the development of HCC.

**Other miRNAs have improved repressiveness without miR-122.** To elucidate how targets of miR-122 and other microRNAs are affected by the loss of miR-122, we performed RNA-seq on mouse liver samples at various stages (21, 45, and 200 days) after

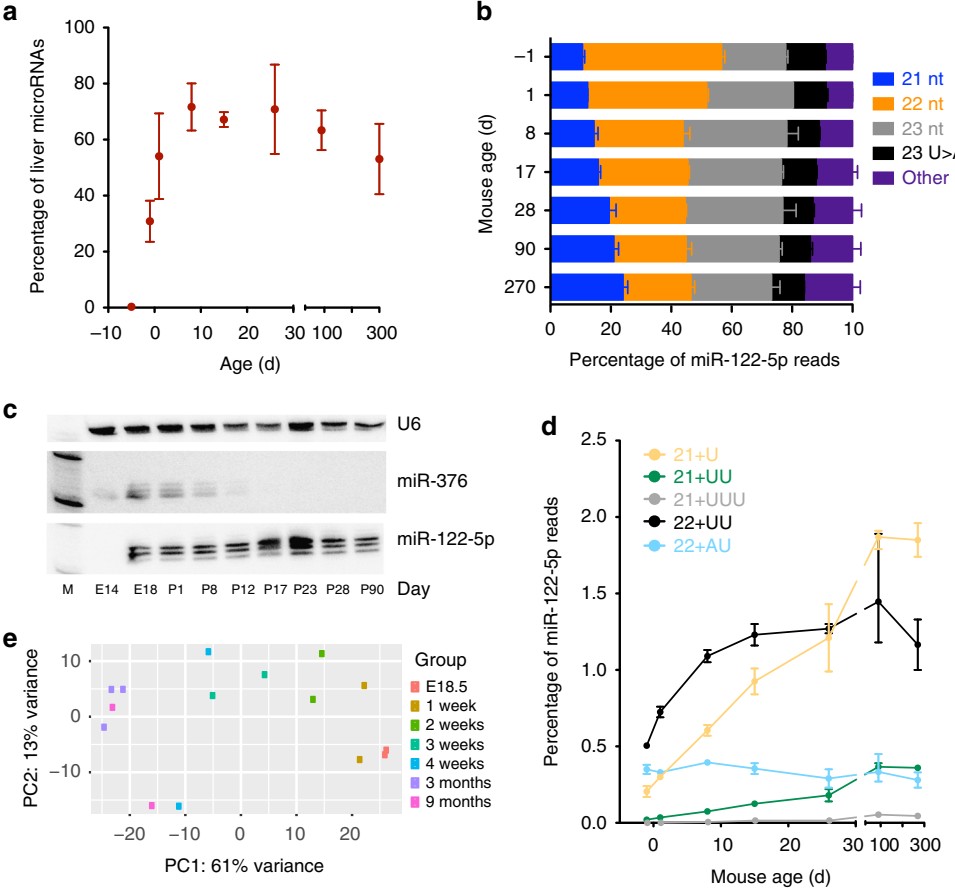

**Fig. 2** Levels of miR-122 and its various isoforms in the developing mouse liver. **a** Percentage of reads that map to miR-122 relative to all microRNAs in mouse embryonic and adult livers ($n = 2$ mice per condition, shown are mean and range). **b** miR-122-5p predominant isoforms at various peri- and postnatal time points. **c** Small RNA northern blot of miR-122 levels. **d** Abundance of rare miR-122-5p isoforms relative to all microRNA reads ($n = 2$ mice per condition, shown are mean and range). **e** Principle component analysis of mouse liver small RNA-seq reads at various ages

Cre-mediated miR-122 removal (Supplementary Data 2). Levels of Cre, as measured by mRNA transcripts from RNA-seq data reveal sustained expression at the 21- and 45-day timepoints and drop-off soon after (Supplementary Figure 4a). The loss of pre-miR-122 was also evident and nearly complete by 21 days (Fig. 4g, h). This corresponded to an increase in the relative expression of the primary long noncoding miR-122 transcript over time (Fig. 4g, h). Finally, loss of the miR-122 precursor region and loss of the ability of Drosha to cleave and terminate transcription, as has been demonstrated previously[19], led to an extension of the primary transcript by 11–12 kb until reaching the adjacent gene, *Alpk2*, that is transcribed in the antisense orientation (Fig. 4g). Whether the long, extended transcript has a function in these mice is unknown. Not surprisingly, miR-122 targets were de-repressed and demonstrated an increase in expression in all evaluated timepoints (Fig. 4a). We additionally found increased abundance of *Mirg* and *Rian*, host genes of the chr12qF1 microRNAs (Supplementary Data 2), indicating both precursor and mature chr12qF1 microRNAs are elevated. Remarkably, we found that in the absence of miR-122, other common microRNAs such as miR-21 could repress their predicted targets to a greater extent than in wild-type livers (Fig. 4b −f), in concordance with data from Argonaute crosslinking and immunoprecipitation sequencing studies in human HCC samples with low miR-122 levels[20]. We validated the RNA-seq results by performing qPCR analysis on the top expressing predicted mRNA target (minimum FPKM of 1) of let-7 and miR-21, *Nr6a1*, *Pdcd4*, *Srsf7* and *Stag2* (Supplementary Figure 4b). Protein levels of Nr6a1 were also decreased starting at the day 14 timepoint

(Supplementary Figure 4c). This leads us to the perhaps expected conclusion that when one common microRNA is removed, other microRNAs can utilize the newly available microRNA machinery to exert a greater repressive effect on their targets.

**Low abundance genes become activated after loss of miR-122**. RNA-seq analysis of liver samples at various timepoints after Cre administration led to increased expression of low-expressed genes. Specifically, the number of genes with an FPKM of 0.1 increased from ~13,000 to ~14,500 by 100 and 200 days after Cre removal (Supplementary Figure 4d). We could show a similar trend using data from an independent study evaluating wild-type and *Mir122* knockout liver mRNAs[20] (Supplementary Figure 4e). By binning our liver RNA-seq samples into quintiles based on their expression, we found that the most highly expressed transcripts were consistently suppressed while more rare transcripts increased in expression (Fig. 4g). This increase was not the result of greater sequencing depth, which was relatively uniform between samples. Instead, the number likely represents the activation of cell types and pathways that are typically silenced by miR-122 and its target genes including transcription factors such as *Cutl1* and *Sox4* [18,21]. Indeed, pathway enrichment analysis revealed genes enriched in cells and tissues such as macrophages, the spleen, thymus, bone marrow, mammary gland, hematopoietic stem cell and lung were significantly upregulated over time after miR-122 removal (Supplementary Figure 5a). As expected, the most downregulated genes were highly enriched in liver expression (Supplementary Figure 5b). Time-course analysis

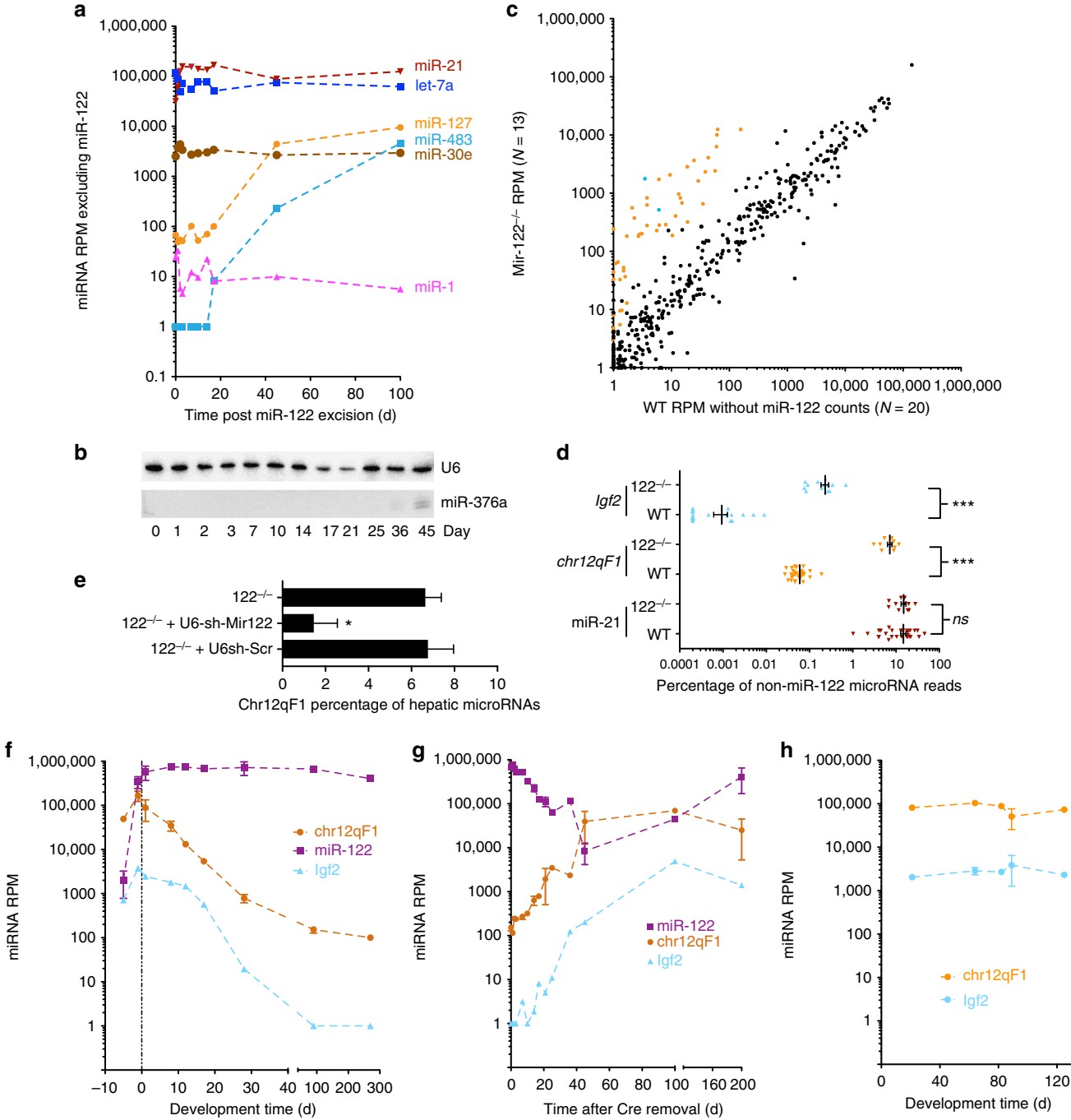

**Fig. 3** Some microRNAs are altered by miR-122 removal, but these are largely in imprinted loci. **a** Reads per million mapped microRNAs (RPM) values for all selected microRNAs at various timepoints after Cre-mediated miR-122 removal including miR-127 at the chr12qF1 *Dlk1-Dio3* locus and miR-483 at the *Igf2* locus (n = 3 for days 0, 7, 14, 21, 45, and 200; n = 1 for remaining timepoints). **b** Small RNA northern blot of miR-376a, present on the *chr12qF1* locus at various points following miR-122 excision. **c** Scatterplot of mean expression per microRNA for small RNA-seq RPM values in miR-122$^{-/-}$ mouse livers relative to wild-type mouse livers. MicroRNAs that map to the chr12qF1 microRNA locus are highlighted in red while those at the *Igf2* locus are blue (n = 13 for miR-122$^{-/-}$ and n = 20 for WT). **d** Comparison of mean expression, expressed as a percentage of total microRNA reads, for miR-21 and the combined normalized read counts for all microRNAs at the chr12qF1 locus (n = 13 for miR-122$^{-/-}$ and n = 28 for WT). **e** Levels of chr12qF1 microRNAs after delivery of an shRNA expressing Mir122 or a control shRNA in miR-122$^{-/-}$ mouse livers (n = 7 mouse livers for 122$^{-/-}$ and 122$^{-/-}$ + U6-sh-Scr, n = 3 mouse livers for 122$^{-/-}$ + U6-sh-miR-122). **f** Expression of miR-122 and all cumulative reads mapping to *chr12qF1* microRNAs at various timepoints during mouse development. **g** Expression of miR-122 and all cumulative reads mapping to *chr12qF1* microRNAs at various timepoints following Cre-mediated miR-122 excision (n = 3 for days 0, 7, 14, 21, 45, and 200; n = 1 for remaining timepoints). **h** Expression of *chr12qF1* and *Igf2* locus microRNAs at various timepoints in miR-122$^{-/-}$ mouse livers

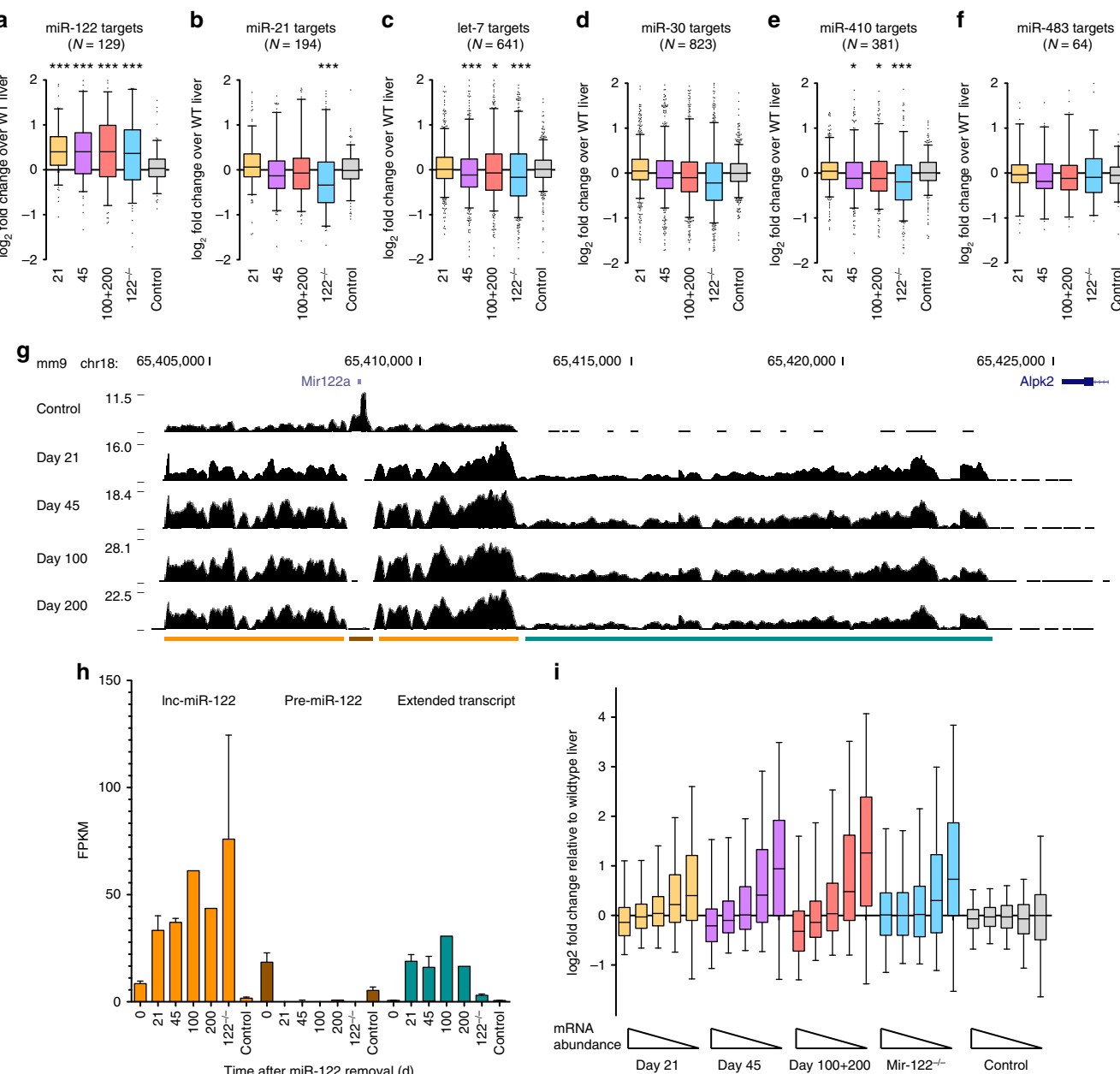

**Fig. 4** Mir122 removal increases miR-122 target expression and reduces targets of other microRNAs. **a**–**f** Box-plot of log₂ fold change expression of predicted targets of microRNAs (*$p < 0.05$; ***$p < 0.001$ by Kruskal–Wallis test followed by Dunn's multiple comparison test, comparing against control). **g** Normalized RNA-seq mapped reads across the long noncoding miR-122 transcript (lnc122). Below is the location of pre-miR-122 (brown), lnc122 (orange) and an extended transcript after miR-122 removal (teal). **h** Quantification of fragments per kilobase of transcript per million mapped reads (FPKM); $n = 3$ per condition with mean ± s.d. except 100 and 200 where $n = 1$. **i** Quintile bins of mRNA expression and deviation from the wild-type liver sample at various timepoints after miR-122 removal

of RNA-seq revealed enrichment in pathways such as focal adhesion, osteoclast differentiation, regulation of actin cytoskeleton and proteoglycans in cancer with strong depletion in metabolic pathways along with the peroxisome and oxidative phosphorylation genes (Supplementary Figure 5c, d). Thus, pathway analysis of RNA-seq data from liver samples devoid of miR-122 reveals the activation of genes typically found in non-liver tissue and activation of specific oncogenic pathways.

## Discussion

In summary, we have identified that when miR-122 expression is abrogated in the adult mouse liver, we can observe a consistent pattern of miR-122-5p isoform degradation. miR-122-5p is processed first through the conversion of the 22-nt isoform to a 21-nt isoform followed by poly-uridylation of the 21-nt isoform to a length of 25 nt or greater. Two other isoforms (23 nt and 23U > A) decay in parallel with the 21-nt and 22-nt isoforms respectively. The half-life of miR-122 based on first-order decay of the time points we have sequenced (Fig. 1a) is 10.2 days. However, considering that the 22-nt isoform of miR-122-5p is synthesized first and also lost first, determining the half-life of the 22-nt isoform level is also relevant—calculated to be 4.3 days. Other miR-122-5p isoforms such as the 21-nt species have a plateau in expression for a few days followed by comparable first-order decay. A considerable effort has been undertaken to accurately count absolute miR-122 levels in the liver, with a recent estimate of 120,000 copies of miR-122 per hepatocyte[1] for this

microRNA that constitutes ~70% of liver microRNAs[13,22,23]. Bearing this in mind, we determine that miR-122 drops from 120,000 copies to 60,000 copies in 10 days after miR-122 excision or one copy of miR-122 is lost every ~15 s. Thus, ~4 copies of miR-122 need to be transcribed per minute to account for miR-122 turnover and maintain a constant level of miR-122 in the liver. Mir-122 has been shown to activate targets in a circadian manner[24] and the requirement of this periodicity may reflect the continual abundant expression.

The absence of miR-122 permits greater suppression of targets of other highly expressed microRNAs. This supports the notion that the number of RISC complexes in a hepatocyte is relatively stable and other microRNAs can vie for empty RISC complexes in the absence of miR-122. The presence of an exogenously delivered shRNA can quell the enhanced repression of other microRNAs when sufficiently expressed and occupying sufficient numbers of RISC complexes. Conversely, delivery of small RNAs compromises the effect of endogenous microRNAs[25], which has implications in other tissues where a few microRNAs or microRNA families account for the majority of microRNA species. In the context of the liver, the loss of this one microRNA has profound effects not only on relative levels of miR-122 targets (which are de-repressed) but also the targets of other highly expressed microRNAs (which are more effectively repressed).

The re-activation of miR-122 after near-complete absence in the liver is also somewhat surprising. Notably, the liver has a propensity to repopulate in conditions of selective pressure[26]. The new miR-122 expression may reflect selective advantage of hepatocytes or other liver cells that did not receive rAAV and have a competitive advantage over ones with an absence of miR-122. Alternatively, if a progenitor-like population within the liver is present and does not take up rAAV-Cre, daughter cells from this population could replenish hepatocytes lost to necrosis/apoptosis. This could explain the presence of chr12qF1 microRNAs which are associated with maintenance of pluripotency[27].

We envision various scenarios for overall microRNA abundance and functional capacity in the liver that could arise from the loss of a predominant microRNA such as miR-122. First, the number of other microRNAs could remain unchanged regardless of the level of miR-122, a "fixed microRNA" scenario. If RISC complexes are in excess, then these microRNAs should exert no further effect on their targets. The other scenario is a "fixed RISC" one whereby the total number of RISC complexes remains constant and as more microRNAs are transcribed and stabilized by RISC. These microRNAs can then suppress their targets to a greater extent. It is more likely that microRNAs are in great excess of RISC complexes so that regardless of the change in absolute microRNA concentration, they can lead to enhanced target suppression in the absence of miR-122. We have also shown strong concordance between total and Ago2-immunoprecipitated small RNA-seq fractions also lending credence to the notion that RISC is limiting[3]. Altogether, however, our data indicate that other microRNAs indeed can act more effectively in the absence of miR-122 (modeled in Supplementary Figure 6).

These data speak to the importance of miR-122 in maintaining repression of targets in the liver but also serving as a buffer to prevent increased repression of mRNAs targeted by other abundant microRNAs. Sustained miR-122 expression is relevant phenotypically as miR-122$^{-/-}$ mice have fibrotic livers and ultimately develop HCC[12,18]. Loss of miR-122 corresponds to the activation of several microRNAs implicated in oncogenesis that are typically highly expressed in embryo development and—through DNA methylation at defined regions—are inactive in the adult. This corresponds to microRNA activation in human HCC which is associated with a stem-like signature and poor survival[28]. Human HCC samples have also revealed the presence of integrated AAV genome fragments at proto-oncogenes with strong evidence that they contribute to tumor progression[29,30]. While no

AAV integration events were detected in the human microRNA cluster or surrounding imprinted locus, an abundance of intergenic variants were detected downstream of miR-122 in the larger noncoding transcript generated after miR-122 excision. The activation of the mouse chr12qF1 locus has been heavily debated with regards to its contribution to the development of HCC[31,32]. The locus is highly expressed following unintended[14,15] or intended[33] targeting of rAAV vectors leading to rAAV vector integration. Mir-122 loss leading to HCC in mice can be tempered by the re-administration of miR-122 by hydrodynamic injection[18]. We are keen to determine which one of the many targets of miR-122 (or secondary effects) lead to chr12qF1 locus activation. However, here we provide definitive evidence of the link between miR-122 loss and chr12qF1 gain, indicating that miR-122 is required to keep the chr12qF1 locus suppressed and prevent HCC development in mice. Importantly, our results demonstrate that miR-122 loss can activate chr12qF1 and Igf2 locus microRNAs rather than preventing their appropriate silencing upon completion of mouse development.

In summary, we have provided a comprehensive analysis of the progression from miR-122 loss to gain of other microRNAs and widespread expression of low abundant mRNA transcripts in the liver. In other tissues, highly expressed microRNAs may also have evolved a dual role of target suppression and buffering of other microRNAs.

## Methods

**Mouse husbandry**. All animal protocols were approved by The Stanford Institute of Medicine Animal Care and Use Committee. A final volume of 200 μl including 5 × 10^11 vector genomes was injected in the tail vein in mice. C57Bl/6 mice were purchased from Jackson Laboratory (Bar Harbor, ME). Mice with loxP sites surrounding miR-122 and miR-122 knockout mice have been previously reported[12]. miR-122 was removed through the delivery of 1×10^12 vector copies (by tail vein injection) of an rAAV expressing Cre under the Ef1a promoter (Addgene plasmid #55636). No animals were excluded from the study and mice were randomly chosen to receive rAAV vectors. Researchers were not blinded to treatment.

**shRNA plasmid design**. Primers for generating a construct expressing miR-122 shRNA are the following: 5′-CACCGAACGCCATTATCACACTCTATACCTGA CCCATATGGGAGTGTGAGAATGGTGTTTTTTGTAC and 5′- AAAAACACCA TTCTCACACTCCATATGGGTCAGGTATAGAGTGTGATAATGGCGTTC as reported previously[3]. AAVs were generated by triple-transfection of pAd5, pAAV8 and various AAV-shRNA constructs as previously reported[34].

**Small RNA deep sequencing**. Mouse liver samples were harvested, snap frozen in liquid nitrogen, ground into powder and extracted using Trizol (Thermo Fisher). Three micrograms of RNA was ligated using T4 RNA ligase in a buffer lacking ATP with a 3′ cloning (NEB) and run on a 15% polyacrylamide gel. A gel fragment corresponding to 17–28 nucleotides of RNA was excised followed by 5′ adaptor ligation incorporating a 4-nucleotide barcode. Small RNA libraries were sequenced on an miSeq machine at the Stanford Functional Genomics Facility (SFGF). After trimming barcodes and adaptors, small RNA reads were aligned to mouse microRNAs from miR-base (release 15)[35] using Bowtie version 0.12.7 [36] allowing for two mismatches.

**RNA sequencing and analysis**. Trizol-extracted RNA samples were poly-A purified using a poly-A spin kit (NEB). Libraries were prepared using a TruSeq kit (Illumina) including barcodes. Sequencing was performed on an Illumina HiSeq 2500 machine at the Stanford Center for Genomics and Personalized Medicine, generating one lane of paired end 101-bp reads. Reads were aligned to the mouse mm9 genome using TopHat (v.2.0.14)[37,38] and to mouse mRNAs to generate FPKM values using Cufflinks v.2.2.1 [37,38] with the following parameters: "–library-norm-method quartile" and removing ribosomal, snoRNA and mitochondrial sequences. Differentially expressed transcripts (false discovery rate < 0.05) were generated using the Cuffdiff program as part of the Cufflinks package.

DAVID v6.7 Gene Ontology program using a Benjamini p value of < 0.05 for significance[39,40]. To calculate the effects of microRNAs on mRNA targets, the list of genes with targets of a given microRNA and their 3′UTR was obtained from Targetscan v.7 [41]. For each mRNA with an FPKM value > 1, the log2-fold change results were plotted. MicroRNA targets and expression values are listed in Supplementary Data 3.

**Cell culture**. Human Embryonic Kidney (HEK) 293 cells were grown in Dulbecco's modified Eagle medium (Gibco-BRL) supplemented with nonessential amino acids,

pen-strep antibiotics, sodium pyruvate, L-glutamine and 10% heat-inactivated fetal bovine serum. Cells in 12-well plates were transfected with 1 µg of various miR-122 constructs using Lipofectamine 2000 and RNA was extracted by Trizol. Cells were obtained from ATCC and were regularly tested for mycoplasma contamination.

**Quantitative PCR**. Probes for *Nr6a1* (catalog Mm00599848), *Stag2* (Mm01311611), *Srsf7* (Mm01233055) and *Pdcd4* (Mm01266062) along with a beta-actin internal control were ordered from Life Technologies for quantitative RT-PCR analysis. Two micrograms of total RNA was reverse-transcribed using a SuperScript II RT kit (Thermo Fisher). Thirty nanograms of cDNA was used for quantitative RT-PCR analysis, run on a CFX384 Real-Time system machine (Bio-Rad) with mouse beta-actin probes used for normalization purposes. Fold change was calculated using the ΔΔ-cT method.

To measure wild-type miR-122 DNA levels, the following forward (5′-TTTGC TAGTGATGGATTGGAAACC) and reverse (5′-AGAGCCCCGGGGATCTTG AATA) primers were used to amplify extracted mouse liver DNA using SYBR-green incorporated dye and measured on a CFX384 Real-Time system machine (Bio-Rad). Values were normalized to input DNA using primers that amplify the beta-actin locus.

**Northern blot studies**. Five micrograms of Trizol-extracted RNA was loaded on an 18% (w/v) acrylamide/7 M urea gel to analyze miR-122 isoform differences at nucleotide resolution and transferred to a Hybond-N + membrane (Amersham). A $^{32}$P-labeled probe complementary to the first 21 nt of miR-122 was used to detect miR-122 isoforms which were then quantified using ImageJ software.

**Western blot studies**. Homogenized mouse liver proteins were lysed in sodium dodecyl sulfate (SDS) lysis buffer (12 mM Tris-Cl, pH 6.8, 2 mM 1,4 dithio-threitol and 1% SDS) with 1× proteinase inhibitor cocktail (Roche). Forty micro-grams of protein samples were loaded on a 4–20% polyacrylamide gel. A mouse anti-Nr6a1 antibody (Abcam, ab38816, 1:1000 dilution) was used to detect the Nr6a1 protein, in comparison to a Gapdh loading control (mouse anti-Gapdh: Sigma-Aldrich, MA5–15738-HRP, 1:50,000 dilution). The protein signals were detected using an Odyssey CLx imaging system (LI-COR Biosciences) according to the manufacturer's instructions. Uncropped blots are shown in Supplementary Figure 7.

**Immunostaining**. Liver tissues were harvested and fixed overnight in 4% wt/vol paraformaldehyde in phosphate buffered saline. Four-micrometersections were cut with a rotating microtome and stained for haematoxylin and eosin or Masson's trichome.

**Statistical analyses**. Data are presented as means ± s.d. or s.e.m. Statistical sig-nificance was tested using GraphPad Prism (version 7). Statistical analysis was performed using Student's *t* test (two-tailed) or one-way ANOVA with Tukey's post-hoc test. A Shapiro−Wilk normality test was performed to ensure Gaussian distribution of samples. For cumulative frequency plots a Kruskal−Wallis test was performed with Dunn's post-hoc multiple correction test against control.

## Data availability

RNA sequencing dataset has been deposited in the NCBI GEO repository with the accession number [https://www.ncbi.nlm.nih.gov/geo/query/acc.cgi?acc=GSE111805]. The remaining datasets are provided in the article and supple-mentary files. All data are available from the authors upon reasonable request.

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

## Acknowledgements

This work was supported by grant NIH R01DK078424 and R01AI071068 (M.A.K.). We are grateful to the Stanford Functional Genomics Facility and Stanford Center for Genomics and Personalized Medicine for high-throughput sequencing services. We would like to thank K. Ghoshal (Ohio State University) for the miR-122$^{-/-}$ mice.

## Author contributions

P.N.V. and M.A.K. conceived the project and designed the experiments and analyzed the data. P.N.V., H.K.K., K.C., F.Z., E.M.M. and J.S. performed the experiments. J.X. provided bioinformatics support. P.N.V. and M.A.K. wrote the manuscript with input from all other co-authors.

## Additional information

**Competing interests:** The authors declare no competing interests.

