## [Peer Review File · Nature Communications]

Reviewers' comments:

Reviewer #1 (Remarks to the Author):

General comments

In the present manuscript Valdmanis et al miR-122 is a highly-expressed liver microRNA that is activated perinatally and aids in regulating cholesterol metabolism and promoting terminal differentiation of hepatocytes.

It is well established that disrupting expression of miR-122 in the liver can re-activate embryo-expressed adult-silenced genes, ultimately leading to the development of hepatocellular carcinoma (HCC). In the present manuscript Valdmanis et al have probed the liver transcriptome at various time points after genomic excision of miR-122 to determine the cellular consequences leading to oncogenesis. The key findings include (1) loss of miR-122 led to specific and progressive increases in expression of imprinted clusters of microRNAs and mRNA transcripts at the *Igf2* and *DIK1-Dio3* loci, (2) these effects could be reversed by re-introduction of exogenous miR-122 and (3) mRNA targets of other abundant hepatic microRNAs became functionally repressed leading to widespread hepatic transcriptional de-regulation.

From these results the authors concluded that data ".... reveals a transcriptomic framework for the hepatic response to loss of miR-122 and the outcome on other microRNAs and their cognate gene targets". These conclusions are well supported by the data presented.

Specific Comments.

In general the experimental designs and data are excellent, and I have no significant comments.

Reviewer #2 (Remarks to the Author):

Valdmanis et al present a series of gene therapy: microRNA studies to examine, in great detail, the biogenesis of miR-122. As the authors state, this microRNA is perhaps the most important hepatic regulator identified to date given the associations with HCC. The design of the studies uses a conditional mouse model of miR-122 and the authors then treat the mice at weaning with an AAV8 configured to express Cre from an EF1 promoter. The kinetics of miR-122 expression is then studied by a version of RNAseq. Expression analysis suggests that loss of miR-122 has severe consequences on many pathways, eventually because non miR-122 microRNAs probably have improved access and production through RISC. The effects are diverse and may drive many pathways that are physiologically relevant. Perhaps most intriguing is the observation that "Loss of miR-122 led to specific and progressive increases in expression of imprinted clusters of microRNAs and mRNA transcripts at the *Igf2* and *DIK1-Dio3* loci that could be curbed by re-introduction of exogenous miR-122." Given that a set of HCCs in humans seems to be strongly associated with this expression pattern, and that in mice AAV mediated HCC seems to also affect these microRNAs, the overall observations here are of great importance for human health and the gene therapy community.

There are a number of areas where the authors should consider expanding or adding more data.

Major areas for commentary.

1. It is interesting and noted by the authors that over time in the AAV treated mice, miR-122 expression reappears. The most obvious explanation is escape from Cre effects (well known with conditional alleles) due to nondeletion. Whether these cells have been further selected to develop uncontrolled growth by an AAV integration even or second hit seems worth considering. More specifically are these cells miR-122 non collapsed hepatocytes on the way toward becoming (or have become) a malignancy? The authors should comment on what the histology revealed, and whether the mice were aged to look at possible susceptibility to HCC? The latter seems very

important to do given the conclusions of the paper. Also, HCC formation in otherwise “resistant” mice can take > 12 months as the original studies of Sands et al clearly demonstrated. Were any of the mice aged? How many alleles at the DNA level were deleted vs non-deleted – this can be determined by a PCR assay over the loxP sites. One would expect proportionally at some level with counts of miR-122 – with more non-collapsed allele associated with higher miR-122 read counts: the expression of miR-122 should be higher, vs a random pattern which might suggest a more complicated genomic event – such as a rearrangement driving the expression of miR-122 with a lower copy number of intact alleles as an alternative explanation for escape from Cre. Could a rogue AAV integration also be a consideration?

2. Why were neonatal AAV studies not performed? The effects might be pronounced in terms of protective vs susceptible to HCC or other pathology.

3. What was the histology in the older mice and can the authors look at Afp levels - higher levels would suggest pre-malignancy perhaps.

4. Is there a place in this work for supportive biochemistry – such as untemplated U addition in vitro to hepatic extracts?

5. Could the mechanism of trans effects on other microRNAs and pathways be further explored – as in western analysis of target genes, or functional studies on hepatocytes, for candidate genes/pathways that are deregulated by loss of miR-122 and its consequences?

Minor:

1. Page 3 talks about rapid AAV expression and is not needed. End at post transduction. AAV8 uncoats and expresses rapidly. This has been well established and comments about this can be deleted entirely from this paper to make room for other edits.

2. Page 6. Isoform patters – patterns?

3. Figure 4 – the predicted targets of microRNAs as discussed in the legend should be listed as a supplemental figure and described in the methods.

4. Methods – add section about miRNA targets, how determined, and list set in sup files.

Reviewer #3 (Remarks to the Author):

In this manuscript, Valdmanis et al describe the effects of Cre/Lox-mediated deletion of the highly expressed liver-specific miR-122 in adult mouse liver. They find that specific miR-122 isoforms change in a pattern that is the opposite of what is seen during liver development. They also identify increases, in other miRNAs expressed from two imprinted clusters as miR-122 decreases, and find that target regulation by other miRNAs in general increases when miR-122 is low. Finally, they show that loss of miR-122 leads to increased expression in generally low expressed, non-liver enriched mRNAs and decreased expression in high expressed, liver-specific mRNAs.

This study makes an important contribution to our understanding of miR-122 function in the adult liver, but there are several weaknesses that need addressing.

1. The title of the paper is perhaps a little misleading as it makes no reference to the miRNAs that increase in abundance in the absence of miR-122, implying that the only change is in miRNA function.

2. The activation of miRNA production from the Igf2 and Chr12qF1 loci following miR-122 removal are very interesting findings. While it is reasonable that the authors have not established a mechanism, it would be straightforward and informative to show qPCR data for the primary transcripts from these loci and confirm that activation is at the level of transcription. It would also be important to show qPCR or northern blot backup for the changes in some of these miRNAs. miR-376 is shown to decrease by northern blot in figure 2c which is important, but the data in figure 3 showing increases in these miRNAs following miR-122 deletion is entirely derived from RNAseq and some confirmation would be useful.

3. The authors observe transcriptional readthrough downstream of the pre-miR-122 which they delete with the Cre/Lox system. This is entirely consistent with published work in which it was

shown that in pri-miR-122 transcription is terminated by Drosha cleavage independent of polyA sites and that transcriptional readthrough occurs when this is inhibited. They reference the paper (reference 18, Dhir et al), but use this reference to suggest that deletion of a polyA site is responsible for the transcriptional readthrough and that the readthrough they have observed is a novel finding, which is inaccurate.

4. Similarly, it would be good to see some reference to Luna et al, Mol Cell 2017, which identifies miR-122 targets in human and mouse liver and identifies some that correlate with HCC survival. Do the gene expression changes that Valdmanis et al suggest contribute to HCC overlap between the two studies?

5. While the legend to figure 4 refers to statistical analysis, there are no asterisks on the figure to indicate whether results are statistically significant. By eye, the decreases described in targets for miRNAs such as let-7 appear very small, and non-existent at d21 at which time miR-122 targets are clearly increased. It would be important to show that these changes are indeed significant to support the authors' conclusion that regulation by other miRNAs increases in the absence of miR-122.

6. The only validation of the changes in miRNA target mRNA expression comes from the qPCR in fig S3b for one target each of let-7 and miR-21. The data in this figure appear very variable and it is not clear how the best fit lines were plotted, as looking at the individual data points it appears more as if the levels of these mRNAs are unchanged over time. At the very least the authors need to show statistical analysis to indicate that there is a genuine decrease in these mRNAs, and ideally also confirm changes in a number of other miRNA targets that they identified in their RNAseq experiments. Given that this observation is the main thrust of the paper, it is very important that it is properly substantiated.

Reviewers' comments:

Reviewer #1 (Remarks to the Author):

General comments

In the present manuscript Valdmanis et al miR-122 is a highly-expressed liver microRNA that is activated perinatally and aids in regulating cholesterol metabolism and promoting terminal differentiation of hepatocytes.

It is well established that disrupting expression of miR-122 in the liver can re-activate embryo-expressed adult-silenced genes, ultimately leading to the development of hepatocellular carcinoma (HCC). In the present manuscript Valdmanis et al have probed the liver transcriptome at various time points after genomic excision of miR-122 to determine the cellular consequences leading to oncogenesis. The key findings include (1) loss of miR-122 led to specific and progressive increases in expression of imprinted clusters of microRNAs and mRNA transcripts at the *Igf2* and *Dlk1-Dio3* loci, (2) these effects could be reversed by re-introduction of exogenous miR-122 and (3) mRNA targets of other abundant hepatic microRNAs became functionally repressed leading to widespread hepatic transcriptional de-regulation. From these results the authors concluded that data “..... reveals a transcriptomic framework for the hepatic response to loss of miR-122 and the outcome on other microRNAs and their cognate gene targets”. These conclusions are well supported by the data presented.

Specific Comments.

In general the experimental designs and data are excellent, and I have no significant comments.

- We very much appreciate the reviewer's enthusiasm about the impact of our work.

Reviewer #2 (Remarks to the Author):

Valdmanis et al present a series of gene therapy: microRNA studies to examine, in great detail, the biogenesis of miR-122. As the authors state, this microRNA is perhaps the most important hepatic regulator identified to date given the associations with HCC. The design of the studies uses a conditional mouse model of miR-122 and the authors then treat the mice at weaning with an AAV8 configured to express Cre from an EF1 promoter. The kinetics of miR-122 expression is then studied by a version of RNAseq. Expression analysis suggests that loss of miR-122 has severe consequences on many pathways, eventually because non miR-122 microRNAs probably have improved access and production through RISC. The effects are diverse and may drive many pathways that are physiologically relevant. Perhaps most intriguing is the observation that “Loss of miR-122 led to specific and progressive increases in expression of imprinted clusters of microRNAs and mRNA transcripts at the *Igf2* and *Dlk1-Dio3* loci that could be curbed by re-introduction of exogenous miR-122.” Given that a set of HCCs in humans seems to be strongly associated with this expression pattern, and that in mice AAV mediated HCC seems to also affect these microRNAs, the overall observations here are of great importance for human health and the gene therapy community.

There are a number of areas where the authors should consider expanding or adding more data.

Major areas for commentary.

1. It is interesting and noted by the authors that over time in the AAV treated mice, miR-122 expression reappears. The most obvious explanation is escape from Cre effects (well known with conditional alleles) due to nondeletion. Whether these cells have been further selected to develop uncontrolled growth by an AAV integration even or second hit seems worth considering. More specifically are these cells miR-122 non collapsed hepatocytes on the way toward becoming (or have become) a malignancy? The authors should comment on what the histology revealed, and whether the mice were aged to look at possible susceptibility to HCC? The latter seems very important to do given the conclusions of the paper. Also, HCC formation in otherwise “resistant” mice can take > 12 months as the original studies of Sands et al clearly demonstrated. Were any of the mice aged? How many alleles at the DNA level were deleted vs

non-deleted – this can be determined by a PCR assay over the loxP sites. One would expect proportionally at some level with counts of miR-122 – with more non-collapsed allele associated with higher miR-122 read counts: the expression of miR-122 should be higher, vs a random pattern which might suggest a more complicated genomic event – such as a rearrangement driving the expression of miR-122 with a lower copy number of intact alleles as an alternative explanation for escape from Cre. Could a rogue AAV integration also be a consideration?

- We appreciate the importance of a qPCR strategy comparing deleted versus non-deleted alleles at LoxP site. To evaluate this strategy as suggested, we designed primers that amplify the wildtype miR-122 allele. qPCR was performed using the various mouse liver samples and revealed that indeed the level of expression of the wildtype allele rebounds in expression, suggesting that the level of miR-122 corresponds roughly with the amount of collapsed and non-collapsed alleles, and we do not have evidence for a more complex genomic re-arrangement. The results have been included as a new supplemental figure 1c.
- We did note that aged mice had tumor-like nodules and succumbed early, though our study was not set up to formally address tumor burden or lifespan changes.
- Regarding rogue AAV integration, the reviewer is likely aware of several instances of AAV integration at the chr12qF1 locus followed by locus activation. To formally exclude this hypothesis as a mechanism for the increased levels of expression that we observe, we mapped RNAseq reads to the AAV genome. This mapping revealed several alignments with paired end reads matching AAV in addition to reads mapping to Cre (supplementary figure 4a). For reads that had only one mapped read we then searched for the mate from the paired-end read in the mouse genome, an approach we were able to successfully employ to identify Adenovirus-Cre integration in mouse lung samples (Valdmanis *et al.*, *Oncogene*, 2015). However, in the current study, only three samples had one read that mapped to the Albumin gene. While this data might suggest AAV integration at Alb as has been demonstrated previously (Chandler *et al.*, *JCI* 2015), no additional junction-spanning reads were present and no integration events were detected at the chr12qF1 or Igf2 loci suggesting that rogue AAV integration was unlikely the cause of our findings.

2. Why were neonatal AAV studies not performed? The effects might be pronounced in terms of protective vs susceptible to HCC or other pathology.

- In general we chose adult mice to have a consistent starting point with respect to miR-122 levels. At earlier time points our results might have been confounded with increasing expression of miR-122 and hepatocyte division which may have led to patchy expression of Cre. We appreciate the idea though and believe it could be interesting for future studies, especially those related to HCC susceptibility. The use of neonates could be useful to expedite tumor generation as a model of disease. Part of our rationale in selecting adults is that miR-122 modulation in neonates will be similar to germline miR-122 knockout mice. By treating adults, we can demonstrate that chr12qF1 and Igf2 locus microRNAs are activated, in contrast to neonatal or germline knockout conditions where we would not be able to distinguish between impaired activation of the locus and inability to silence the locus. This distinction is actually an important point and we are glad this reviewer prompted us to elaborate. We have added a sentence in the discussion to reflect this point: "Importantly, our results demonstrate that miR-122 loss activates *chr12qF1* and *Igf2* locus microRNAs rather than preventing their appropriate silencing during development."

3. What was the histology in the older mice and can the authors look at Afp levels - higher levels would suggest pre-malignancy perhaps.

- To address this important comment from the reviewer, we performed histology on older mice. Generally, the results revealed a milder pattern of pathology than what was observed in germline knockout or liver-specific knockout of miR-122. However, we did observe evidence of microsteatosis and fibrosis, the latter of which was confirmed by Masson's trichrome staining though at the time we did not have additional sections available to stain for AFP. The details are now included as a new supplemental figure (see below), and we have updated the text as follows

in the results section: “Immunohistochemistry of mouse liver sections 200 days after rAAV-Cre indicates fibrosis as has been described for miR-122 knockout mice (Supplementary Fig. 3)”

4. Is there a place in this work for supportive biochemistry – such as untemplated U addition in vitro to hepatic extracts?

- This is a valid consideration, though we believe, while it may provide complementary findings the experiments likely will not yield novel information and the main conclusions not already captured in the manuscript and considering the technical challenges with establishing such a protocol we believe they will be better suited for follow-up reports.

5. Could the mechanism of trans effects on other microRNAs and pathways be further explored – as in western analysis of target genes, or functional studies on hepatocytes, for candidate genes/pathways that are deregulated by loss of miR-122 and its consequences?

- Western blots are indeed a valuable experimental technique to confirm consequences of microRNAs that are de-repressed. To explore this line of experimentation, we performed a western blot on the Nr6a1 protein, whose mRNA is a target of the let-7 microRNA. These results revealed that indeed protein levels of Nr6a1 decrease upon miR-122 removal but, like the level of miR-122 itself, start to recover in terms of expression by day 200. We were unable to detect Stag using an antibody against this protein. The Nr6a1 results are included as a new supplemental figure 4c (see below).

Minor:

1. Page 3 talks about rapid AAV expression and is not needed. End at post transduction. AAV8 uncoats and expresses rapidly. This has been well established and comments about this can be deleted entirely from this paper to make room for other edits.

- We appreciate this comment and have modified the manuscript as suggested.

2. Page 6. Isoform patters – patterns?

- Yes, thank you for picking up on this typo, we have updated the manuscript.

3. Figure 4 – the predicted targets of microRNAs as discussed in the legend should be listed as a supplemental figure and described in the methods.

- We have added the predicted targets as a new supplemental table 3. We have also updated the methods section.

4. Methods – add section about miRNA targets, how determined, and list set in sup files.

- In addition to the new supplemental table 3, we have provided a more detailed section in the methods describing the miRNA targets. Specifically, Targetscan v7 was used to identify all mouse targets of a given microRNA and the log2 fold change of each was calculated. The methods have been updated as follows: “For each mRNA with an FPKM value >1, the log2-fold change results were plotted. MicroRNA targets and expression values are listed in Supplemental Table 3.”

Reviewer #3 (Remarks to the Author):

In this manuscript, Valdmanis et al describe the effects of Cre/Lox-mediated deletion of the highly expressed liver-specific miR-122 in adult mouse liver. They find that specific miR-122 isoforms change in a pattern that is the opposite of what is seen during liver development. They also identify increases, in other miRNAs expressed from two imprinted clusters as miR-122 decreases, and find that target regulation by other miRNAs in general increases when miR-122 is low. Finally, they show that loss of miR-122 leads to increased expression in generally low expressed, non-liver enriched mRNAs and decreased expression in high expressed, liver-specific mRNAs.

This study makes an important contribution to our understanding of miR-122 function in the adult liver, but there are several weaknesses that need addressing.

1. The title of the paper is perhaps a little misleading as it makes no reference to the miRNAs that increase in abundance in the absence of miR-122, implying that the only change is in miRNA function.

- We agree that the title can better reflect our findings and have changed it to “Removal of miR-122 in the liver activates imprinted microRNAs and enables more effective microRNA-mediated gene repression”

2. The activation of miRNA production from the Igf2 and Chr12qF1 loci following miR-122 removal are very interesting findings. While it is reasonable that the authors have not established a mechanism, it would be straightforward and informative to show qPCR data for the primary transcripts from these loci and confirm that activation is at the level of transcription. It would also be important to show qPCR or northern blot backup for the changes in some of these miRNAs. miR-376 is shown to decrease by northern blot in figure 2c which is important, but the data in figure 3 showing increases in these miRNAs following miR-122 deletion is entirely derived from RNAseq and some confirmation would be useful.

- We have generated an additional northern blot indicating the slow rise of miR-376 which is more readily detectable by high throughput sequencing but does appear on northern blots 200 days after Cre removal of miR-122. This blot now appears as a new figure 3b.
- We also took advantage of RNA-seq data to evaluate read depth across the *Mirg* and *Rian* gene on chr12qF1, as an indicator of the read depth of the primary transcript of this locus, since the microRNAs are processed from these two host genes. Normalized (FPKM) RNA-seq read counts revealed a drastic upregulation in all time points relative to day zero and in the miR-122 knockout mice. The data is pasted below and we also included this information in the text: "We additionally found increased abundance of *Mirg* and *Rian*, host genes of the chr12qF1 microRNAs (Supplementary Table 2) indicating both precursor and mature chr12qF1 microRNAs are elevated."

Sample	Rian	Mirg
miR122_flox_d0_repA	0.236	0.015
miR122_flox_d0_repB	0.123	0.000
miR122_flox_d0_repC	0.234	0.061
miR122_floxCre_d21_repA	0.952	0.169
miR122_floxCre_d21_repB	1.867	0.388
miR122_floxCre_d21_repC	5.803	1.602
miR122_floxCre_d45_repA	12.008	3.564
miR122_floxCre_d45_repB	45.564	12.421
miR122_floxCre_d45_repC	38.886	8.452
miR122_floxCre_d100	50.780	11.504
miR122_floxCre_d200	29.031	5.890
122ko_repA	11.158	6.977
122ko_repB	26.109	7.397
122ko_repC	40.545	9.185

3. The authors observe transcriptional readthrough downstream of the pre-miR-122 which they delete with the Cre/Lox system. This is entirely consistent with published work in which it was shown that in pri-miR-122 transcription is terminated by Drosha cleavage independent of polyA sites and that transcriptional readthrough occurs when this is inhibited. They reference the paper (reference 18, Dhir et al), but use this reference to suggest that deletion of a polyA site is responsible for the transcriptional readthrough and that the readthrough they have observed is a novel finding, which is inaccurate.

- We thank the reviewer for pointing out our error in interpretation, we inadvertently confused details in this manuscript with a related presentation. We have altered the text to reflect the role of Drosha in pri-miR-122 transcription processing, and appreciate again that the reviewer informed us of this inaccuracy.

4. Similarly, it would be good to see some reference to Luna et al, Mol Cell 2017, which identifies miR-122 targets in human and mouse liver and identifies some that correlate with HCC survival. Do the gene expression changes that Valdmanis et al suggest contribute to HCC overlap between the two studies?

- We have evaluated the targets from the Luna *et al* manuscript as a new series of analyses in the current manuscript. The results indicate that similar to our analyses, the number of transcripts that have marginal expression (FPKM range of 0.1-1) increases in miR-122 knockout mice as shown in a new supplemental figure 4d. We have also added the reference to Luna et al in the

main text to reference their identification that Argonaute CLIP identified peaks with miR-21 binding sites are suppressed in human HCC samples.

5. While the legend to figure 4 refers to statistical analysis, there are no asterisks on the figure to indicate whether results are statistically significant. By eye, the decreases described in targets for miRNAs such as let-7 appear very small, and non-existent at d21 at which time miR-122 targets are clearly increased. It would be important to show that these changes are indeed significant to support the authors' conclusion that regulation by other miRNAs increases in the absence of miR-122.

- We thank the reviewer for pointing out the missing significance values which we had unfortunately omitted during figure generation and recognized only after submission. We have now gone back and added back the values. As the reviewer correctly points out, the decrease in targets of all tested microRNAs is not significant at day 21 but is present at day 45 for some but not all microRNAs. In addition, in consulting with others we realize that our statistical test was not optimal given that our data do not follow a Poisson distribution and have thus instead used a Kruskal-Wallis test followed by Dunn's multiple comparison.

6. The only validation of the changes in miRNA target mRNA expression comes from the qPCR in fig S3b for one target each of let-7 and miR-21. The data in this figure appear very variable and it is not clear how the best fit lines were plotted, as looking at the individual data points it appears more as if the levels of these mRNAs are unchanged over time. At the very least the authors need to show statistical analysis to indicate that there is a genuine decrease in these mRNAs, and ideally also confirm changes in a number of other miRNA targets that they identified in their RNAseq experiments. Given that this observation is the main thrust of the paper, it is very important that it is properly substantiated.

- We agree that the qPCR data is not very compelling and have gone back to re-extract RNA and perform additional replicates, focusing on the samples for which we have biological triplicates (days 7,14,21,45 and 200). The linear regression we had performed we agree is not convincing as the underlying data is quite variable. We have instead re-plotted the data as a bar chart to match side-by-side panels in supplemental figure 4. We also selected two additional well-expressed targets of common microRNAs, *Srsf7* and *Pdcd4*. We then performed a one-way ANOVA for each sample to demonstrate that all four targets were significantly decreased in miR-122 depleted livers at various points though the expression recovered at day 200. We note that for several genes, additional mechanisms are in play for preserving expression and thus it is challenging and time consuming to demonstrate on a gene-by-gene basis, pointing to the importance of transcriptome-wide analysis of microRNA targets. In addition, we would like to point out that we have performed western blot analysis of the *Nr6a1* target (see response to reviewer 2 point 5) which more accurately reflects its relative abundance in hepatocytes and have found that protein levels are also decreased with respect to the wildtype liver.
- See new data below:

REVIEWERS' COMMENTS:

Reviewer #2 (Remarks to the Author):

A revised manuscript is submitted, and while not all comments have been addressed or fully explored (example : treating neonates vs adults and then aging the mice to study the effects), the authors have performed a fair number of new experiments and the paper has been greatly improved.

Reviewer #3 (Remarks to the Author):

I am satisfied with the authors' response to my original comments.